# Avoiding siloed approaches: Integrating psychological insights into sustainable farming

John Maltby *

School of Psychology and Vision Sciences, University of Leicester, Leicester, Leicestershire, United Kingdom

* jm148@leicester.ac.uk

**Data Availability Statement:** The data can be found at the Center for Open Science https://osf.io/9kuhq/.

**Funding:** JM EP/Y00597X/1 Engineering and Physical Sciences Research Council https://www.ukri.org/councils/epsrc/ The funders did not play

## Abstract

This study enhances our understanding of the psychological factors influencing farmers' adoption of sustainable farming practices, specifically those related to achieving NetZero emissions. It achieves this by integrating various psychological theories with practical farming methods within the context of Behavioral-Adoption and Purpose-Driven contexts. The research consisted of two studies. Study 1 employed Exploratory Factor Analysis (EFA) to analyze responses from 438 UK farmers regarding their attitudes toward a series of Net Zero policy commitments, drawing on psychological theories including the Unified Theory of Acceptance and Use of Technology, the Theory of Planned Behavior, the Prototype Willingness Model, Implementation Intentions, Self-Determination Theory, Eudaimonia, and the Integrated Model of Health Literacy. The findings revealed a new Integrated Motivation Model for Sustainable Farming that comprises seven factors: Agricultural Commitment and Stewardship, Sustainable Farming Readiness and Confidence, Sustainable Incentive Engagement and Acceptance, Climate Adaptation Competence and Confidence, Net Zero Accountability and Reporting Commitment, Community Influence and Commitment in Sustainable Farming, and Innovation and Technological Competence. Study 2 validated these factors through the development of a 21-item Integrated Motivation Model for Sustainable Farming scale and use of Confirmatory Factor Analysis (CFA) to confirm the 7-factor structure using a subsample of 418 UK farmers from Study 1 and an additional 230 US farmers. Furthermore, Study 2 tested the concurrent validity of the new scale by demonstrating significant associations with reported sustainable farming behaviors. These findings underscore the complex interplay of motivational, cognitive, and social processes influencing sustainable farming practices. The integrated psychological model developed through this research provides parsimonious and valuable insights for policymakers to foster sustainable practices in farming effectively. The confirmation of this model across farming populations enhances its generalizability and potential to guide targeted interventions aimed at achieving behavioral change in pursuit of Net Zero targets in agriculture.

any role in the study design, data collection and analysis, decision to publish, or preparation of the manuscript.

**Competing interests:** The authors have declared that no competing interests exist.

## Introduction

Confronting climate change demands an urgent shift towards net-zero emissions, spotlighting agriculture's critical role in this global endeavor. The sector's transformative potential is underscored by the necessity of integrating psychological insights to navigate and expedite this transition effectively. Farmers and land managers are at the forefront, adopting sustainable farming practices that not only reduce their carbon footprint but also contribute significantly to a more sustainable future. These efforts are supported by a growing body of psychological research that offers valuable insights into the dynamics of adopting net-zero policies and sustainable technologies within the agricultural community.

In this consideration, we present two main psychological contexts that influence farmers' decision-making towards sustainable farming: Behavioral-Adoption and Purpose-Driven frameworks. The Behavioral-Adoption frameworks, often cited in sustainable farming literature, focus on external factors like social influence and practical utility, which are crucial for the adoption of new technologies and practices by farmers. For instance, these frameworks explore how peer attitudes and the availability of supportive conditions can encourage sustainable behaviors. On the other hand, the Purpose-Driven frameworks delve into the internal motivations of farmers, such as personal fulfillment and alignment with core values, which drive their engagement in sustainable practices. These frameworks emphasize understanding why farmers choose to engage in sustainability from an intrinsic perspective. By contrasting and integrating these frameworks, our study offers a comprehensive view of the dynamic interplay between external influences and internal motivations that shape sustainable agricultural practices.

In terms of how these two sets of frameworks relate to specific psychological models and sustainable farming, the first set to consider are Behavioral-Adoption frameworks. An exampler model is the Unified Theory of Acceptance and Use of Technology [1], which explains technology acceptance by highlighting factors like performance expectancy (the technology's perceived usefulness), social influence (the impact of peer attitudes), and facilitating conditions (the presence of a supportive environment). It is these elements of the Unified Theory of Acceptance and Use of Technology model that significantly influences farmers' attitudes toward adopting sustainable farming practices [2]. Complementing the Unified Theory of Acceptance and Use of Technology, the Theory of Planned Behavior [3] provides a framework around adopting sustainable technologies in farming via intentions, shaped by attitudes, norms, and perceived control, predict behaviors. Research shows that attitudes, social norms, and perceived control positively influence farmers' intentions to adopt practices like organic farming and improved natural grassland management [4–6].

The second set to consider are Purpose-Driven frameworks that focus on beliefs, meaning, fulfillment, desired prototypes, environmental literacy processes, and contributions to personal and community well-being. Within the existing sustainable farming literature, Self-Determination Theory (SDT; [7]) has been used to explore practices in sustainable farming as reflections of purpose-driven intentions in farming, in terms of motivational intrinsic versus extrinsic needs driven by autonomy, competence, and relatedness. Research shows that SDT principles are instrumental in how a self-directed approach contributes to sustainable farming, emphasizing how intrinsic choices over external rewards and fostering conditions that support farmer autonomy leads to sustainable farming [8, 9]. Moreover, integrating models such as the Prototype Willingness Model [10] and the theory of Implementation Intentions [11, 12] could offer an advanced understanding of concepts of prototypical influences around the identity of being a sustainable farmer (e.g., the 'eco-friendly farmer') and intentional, planned behaviors (e.g. specific plans) in sustainable farming are Purpose-Driven factors in farming. For example,

in terms of implementation intentions, developing detailed action plans for pesticide reduction or water conservation can underline the practical applicability of these theories [13, 14]. There are also other psychological frameworks that we could consider Purpose-Driven. The inclusion of models of environmental literacy [15] that enriches our understanding by framing the competencies necessary for sustainable farming. This approach not only emphasises individuals' abilities to access and understand information relating to sustainable farming, but also emphasis the ability to make purposeful and well-informed decisions that positively impact on sustainable farming. Lastly, though there has been an exploration of the relationship between everyday affect (e.g. enjoyment) and farming practices [16, 17] there is a compelling need for a broader examination of well-being concepts, particularly eudaimonia. Eudaimonia focuses on fulfilling one's potential and finding deep meaning and happiness in alignment with one's true self [18, 19]. Exploring how eudaimonia correlates with sustainable farming practices [20] offers vital insights into possibly why fulfilment and meaning may be purposefully connected with the adoption of sustainable practices in farming.

These two broad psychological frameworks, Behavioral-Adoption and Purpose-Driven, support the psychological, motivational, cognitive, and social processes that influence farmers' choices towards sustainable farming. Although the wealth of models [1, 3, 7] offers deep insights into the adoption of sustainable practices, their complexity can hinder the creation of straightforward, unified strategies. By identifying common themes across these frameworks, we can simplify complex theories into practical strategies that prevent siloed approaches (an approach where different sectors or disciplines operate in isolation) and enhance farmers' and land managers' adoption of sustainable practices. This integration helps bridge the gap between intricate psychological theories and farming realities, promoting a more cohesive understanding. For example, in regions where farming demands significant water resources, leading to environmental stress, especially during drought conditions, promoting water-efficient technologies poses a challenge [21]. Farmers may be hesitant due to high upfront costs, uncertainty about effectiveness, and disruption of established practices. A better understanding can be achieved by integrating factors from Behavioral-Adoption and Purpose-Driven frameworks. We could identify factors that emphasize practical readiness and external factors such as financial incentives (Behavioral-Adoption), which align with farmers' values towards sustainability and stewardship of natural resources (Purpose-Driven). This integration provides a more meaningful and strategic understanding of these factors. Similarly, in areas where farming is crucial to the economy and vulnerable to climate change [22], a unified model that integrates Behavioral-Adoption strategies—demonstrating the effectiveness and efficiency of new sustainable practices—with the Purpose-Driven aspects that engage farmers' values and cultural practices related to land and community welfare, can be advantageous. This dual approach ensures farmers are not only prepared but also motivated to adopt new practices that benefit both their livelihood and their community. Thus, this unified approach addresses both the logistical and motivational aspects of adopting or changing behaviors to target sustainable farming.

Integrating psychological frameworks, however, presents certain challenges. A primary concern is the potential conflict between theoretical assumptions. Behavioral-Adoption frameworks, for example, often emphasize external motivations like incentives or regulatory pressures, which may contradict the intrinsic motivations highlighted by Purpose-Driven frameworks. This conflict can create tension in designing interventions that need to balance immediate benefits with long-term sustainable goals. Additionally, there are methodological challenges related to scale compatibility and data integration when merging the Behavioral-Adoption and Purpose-Driven frameworks. These frameworks not only originate from different theoretical backgrounds but also use distinct scales and measurement tools that may not

directly align. For instance, some theories such as Eudaimonia [19] and Self-Determination Theory [23] feature psychometrically validated scales providing clear, measurable indicators. Others, like the Unified Theory of Acceptance and Use of Technology [1] and health literacy models [24], require scales specifically adapted to the behavior under study. To address these issues, we propose adopting the abductive method [25]—a reasoning approach that begins with observations and seeks the simplest, most likely explanation. This leads us to suggest developing a new model that integrates diverse psychological factors, focusing specifically on sustainable farming within the NetZero context. This model would incorporate new constructs along with measures that capture the motivational, cognitive, and social processes influencing farmers. By synthesizing diverse models into a unified one, we aim to move beyond siloed approaches and create an integrated framework that effectively promotes sustainable practices in farming.

The primary aim of this study is to develop a comprehensive psychological model that merges the Behavioral-Adoption and Purpose-Driven frameworks around sustainable farming practices to achieve NetZero among farmers. This integrated model will enhance our understanding of the psychological factors that influence farmers' behaviors and motivations towards sustainable farming practices to achieve NetZero goals. Specifically, there are three objectives:

1. Develop an integrated psychological framework that integrates diverse perspectives to explain behavior change in the context of sustainable farming. This framework will utilize exploratory factor analysis (EFA) as an abductive reasoning method [26] to identify underlying psychological factors from a broad array of existing psychological models and constructs that reflect Behavioral-Adoption and Purpose-Driven frameworks.

2. Demonstrate structural validity for the proposed integrated psychological model through confirmatory factor analysis (CFA) [27], ensuring that our model reliably captures the essential motivational, cognitive, and social processes that influence farmers' decisions.

3. Assess the model's practical implications by linking the psychological factors to actual behavioral indicators of NetZero farming practices, thereby demonstrating the model's concurrent validity.

## Study 1

Study 1 was designed to explore, using Exploratory Factor Analysis, the creation of an integrated psychological model of attitudes towards six factors of Net Zero policy commitments (Emission Reduction Targets, Regulatory Frameworks, Financial Incentives, Reporting and Verification, Innovation and Support, and Adaptation and Resilience). This model utilizes the following psychological domains: the Unified Theory of Acceptance and Use of Technology, the Theory of Planned Behavior, the Prototype Willingness Model, Implementation Intentions, Eudaimonia, Self-Determination Theory, and the Model of Health Literacy. Synthesising these theories offers a comprehensive framework to explore the factors that drive motivation, planning, fulfillment, and literacy drive in sustainable farming.

### Method

For both studies consent was facilitated through an online survey, where respondents indicated their consent by affirmatively selecting the consent option on the digital form confirming consent statements. This consent option confirmed that participants were fully informed and voluntarily agreed to participate in the study.

**Table 1. Demographic variables for the three samples for Study 1 and Study 2.**

|  |  | Study 1 | Study 2 | |
|---|---|---|---|---|
|  |  | UK sample | UK sample* | USA sample |
| Age (Mean, SD) |  | 32.70 (10.99) | 32.63 (10.93) | 32.85 (11.58) |
| Years Experience (Mean, SD) |  | 7.14 (7.34) | 7.08 (7.24) | 7.80 (7.51) |
|  | Category |  |  |  |
| Gender | Male | 272 | 261 | 137 |
|  | Female | 165 | 156 | 93 |
|  | Other | 1 | 1 | 0 |
| Role | Sole Manager | 83 | 79 | 25 |
|  | Part of Management Team | 142 | 134 | 48 |
|  | Employee with management responsibilities | 111 | 106 | 75 |
|  | No management responsibilities | 86 | 83 | 62 |
|  | Other | 16 | 16 | 20 |
| Farm Size | 2 hectares or less | 93 | 87 | 62 |
|  | 2 hectares to 10 hectares | 136 | 130 | 72 |
|  | 10–50 hectares | 107 | 101 | 45 |
|  | 50–100 hectares | 54 | 53 | 24 |
|  | Over 100 hectares | 48 | 47 | 27 |
| Farm Type | Arable | 274 | 258 | 133 |
|  | Livestock | 118 | 116 | 30 |
|  | Arable/Livestock | 46 | 44 | 67 |

Key

* Sample is a subsample of the Study 1 sample

**Sample.** The sample for Study 1 included 438 UK farmers. Recruitment was conducted through the online crowdsourcing platform Prolific, utilizing two methods to reach potential participants. The first method targeted individuals in the UK who were registered on the site as working in the agricultural industry, identified through a screening question regarding employment in the 'Agriculture, Food, and Natural Resources' sector, ensuring a relevant participant pool. The second method invited individuals who self-identified as farmers to register for a short screening study. This was necessary because those on the crowd-sourcing site might not have originally registered their employment sector with the crowdsourcing platform due to reasons such as lack of perceived relevance, privacy concerns, changes in role, or misalignment with professional identity. Subsequently, those who were not already identified via the first method were invited to the main study via the crowdsourcing platform, thereby broadening the potential respondent base. The gender distribution within the sample was 272 males, 165 females, and 1 individual who did not identify with either category. The mean age of the participants was 32.5 years, with a standard deviation of 11.02 years. Other demographic information for this sample is provided in Table 1. The recruitment period for participants was 12th December 2023 – 12th January 2024. In alignment with ethical guidelines, informed consent was obtained from all participants. This consent was facilitated through an online survey, where respondents indicated their consent by affirmatively selecting the consent option on the digital form confirming consent statements. This consent option confirmed that participants were fully informed and voluntarily agreed to participate in the study.

**Questionnaire.** An original questionnaire was developed to explore the intersection between each Net Zero component and relevant psychological factors. At the center of this work is the integration of five broader psychological frameworks, including (1) the Unified

Theory of Acceptance and Use of Technology [1], (2) a combination of theories such as the Theory of Planned Behavior [3] alongside concurrent concepts like the Prototype-Willingness Model [10] and Implementation Intentions [11, 28], with (3) theories of health literacy [24], (4) Eudaimonia [19], and (5) Self-Determination Theory [7]. The utilization of these theories provides a comprehensive narrative for understanding and fostering farmer Behavioral-Adoption and Purpose-Driven sustainable practices around NetZero.

First, the Unified Theory of Acceptance and Use of Technology underscores seven factors: (1) Performance Expectancy, where farmers recognize the sustainability improvements from adopting NetZero; (2) Effort Expectancy, which suggests that these technologies should be manageable and straightforward; (3) Social Influence, highlighting the role of peers and the agricultural community in shaping adoption decisions. Additionally, (4) Facilitating Conditions, (5) Hedonic Motivation, (6) Price Value, and (7) Habit factor emphasize the need for accessible resources, personal satisfaction, awareness of long-term benefits, and integrating sustainable farming into routine management strategies, respectively.

Second, the theories of Planned Behavior, Implementation Intentions, and Prototype Willingness Model extend consideration to eight more factors, focusing on Attitude toward the Behavior, where a positive mindset towards sustainable farming is essential. Subjective Norm and Perceived Behavioral Control reflect the community's expectations and the farmer's confidence in their abilities, respectively. The theories also emphasize Willingness to adapt, Prototypes of ideal farming behavior, Specificity of Plan for action, Initiation of the b, and having Coping Plans to address challenges in NetZero.

Third, to assess literacy in this context, we adapted a Health Literacy Model (24) that suggests four factors to determine whether farmers can (1) access/obtain, (2) understand, (3) process/appraise, and (4) apply/use information efficiently, enabling informed decision-making in their operations around sustainable farming.

Fourth, Eudaimonia covers six factors that reflect deeper well-being motivations, with practices that foster Self-Acceptance, Personal Growth, Purpose in Life, Environmental Mastery, Autonomy, and Positive Relations with Others, aligning farming around NetZero with personal values and community well-being.

Finally, Self-determination Theory emphasizes three main factors: Autonomy in decision-making, Competence in implementing sustainable strategies effectively, and Relatedness, reinforcing the connection with the community through shared sustainable efforts.

A summary of these five broad approaches–and the subsequent 28 factors–is summarized in Table 2.

We applied the 28 psychological factors to explore the domain of sustainable farming, each to six aspects of Net Zero policy commitments [29–33]. First, Emission Reduction Targets: Agriculture significantly contributes to global greenhouse gas (GHG) emissions. Transitioning towards cleaner, less emission-intensive energy systems and agricultural practices is crucial. Strategies such as promoting plant-based diets and reducing livestock numbers are key to lowering emissions. Second, Regulatory Frameworks: The transition to net zero in agriculture necessitates comprehensive policy shifts. This includes redirecting public support towards environmental public goods, setting incentives for farmers to adopt low-emission practices, and encouraging dietary changes towards consuming less meat. Third, Financial Incentives: The agricultural sector receives substantial public financial support globally. Redirecting this support towards sustainable practices could significantly influence emission reductions and environmental sustainability. Public support mechanisms need reforming to align with climate goals, suggesting a move from production-based to decoupled payments that reward environmental stewardship. Fourth, Reporting and Verification: The importance of monitoring, reporting, and verification frameworks is underscored by the need for accurate accounting of

**Table 2. Psychological models and components used in the current study.**

| Theory | Components | Illustration of Behavior |
|---|---|---|
| Unified Theory of Acceptance and Use of Technology | Performance Expectancy | Possible improvement/enhancement |
| | Effort Expectancy | Manageable and straightforward |
| | Social Influence | Influenced by peers and agricultural community |
| | Facilitating Conditions | Necessary resources, information |
| | Hedonic Motivation | Personal satisfaction |
| | Price Value | Long-term benefits of investing |
| | Habit | A routine part of the strategy. |
| Theory of Planned Behavior, Prototype Willingness Model, Implementation Intentions. | Attitude | Positive attitudes towards the behavior |
| | Subjective Norm | Sense of responsibility from wider community |
| | Perceived Behavioral Control | Confident to implement |
| | Willingness | Open to adapting practices |
| | Prototype | Seen as a prototypical of being responsible and innovative farmer |
| | Specificity of Plan | Clear and detailed plan |
| | Behavior Initiation | Ready to start or enhance practices |
| | Coping Plans | Strategies in place to address challenges |
| Literacy | Access/Obtain Information | Easily access information |
| | Understand Information | Understand the concepts and practices |
| | Process/Appraise Information | Capable of evaluating and applying information |
| | Apply/Use Information | Use available information to make farming operations decisions |
| Eudaimonia | Self-Acceptance | Accepting positive and limitations of self |
| | Personal Growth | Contributes to growth |
| | Purpose in Life | Purpose and direction |
| | Environmental Mastery | Capable of effectively managing environment |
| | Autonomy | Independently chooses to implement practices |
| | Positive Relations with Others | Positive relationships within community |
| Self-Determination Theory | Autonomy | Autonomously engage in practices as part of decision-making |
| | Competence | Feel skilled and effective in implementing strategies |
| | Relatedness | Efforts strengthen wider connection |

methane emissions. Given methane's short-lived nature and significant impact on global warming, refined measurements are essential to ensure the effectiveness of reduction strategies. Fifth, Innovation and Support: Improvements in farming efficiency, such as enhancing animal health, using precision farming, and adopting energy-efficient measures, are vital. Innovations like low-methane feed additives and improved manure management can significantly reduce methane and nitrous oxide emissions. Sixth, Adaptation and Resilience: Strategies to enhance carbon capture, along with adopting low-carbon farming techniques and reducing food waste, not only mitigate emissions but also improve the resilience of agricultural systems against climate impacts.

The questionnaire was structured into six distinct sections, each corresponding to one of the Net Zero Policy Commitment criteria. Within each section, there were 28 factors to consider, leading to a comprehensive collection of 168 factors (or indicators) across the questionnaire. These indicators are detailed in S1 Table and were presented to participants in a

random order. Each item on the questionnaire was evaluated using a Likert-type scale, which ranged from 1, indicating strong disagreement, to 5, signifying strong agreement.

## Results

The initial phase of our analysis involved exploring the factor structure of the questionnaire items using SPSS for Windows 28. Given the inclusion of numerous items, some of which might not directly relate to our proposed constructs, we utilized Exploratory Factor Analysis (EFA) to allow for the organic emergence of any factor structure. Our EFA included a scale of 168 items with 438 participants and acknowledged that the sample size did not meet the ideal subject-to-variable ratio of 5 participants for every 1 item. However, this was deemed acceptable for several reasons. First, with a sample size classified as 'fair' to 'good' according to [34], and our EFA's adequacy was affirmed, as indicated by a Kaiser-Meyer-Olkin (KMO) measure of 0.966. A KMO value significantly above the 0.8 threshold indicates the suitability of data for factor analysis [34, 35]. Furthermore, the practical constraints often present in specialized research fields, such as surveying UK farmers, necessitate a balance between statistical ideals and feasibility [36]. Therefore, the adequacy and suitability could be confirmed in this context.

Parallel analysis, recommended as the most accurate method for determining the number of factors by various reports [37–39], was employed in our study. The 8th eigenvalue (64.293, 10.154, 6.297, 4.116, 3.110, 2.973, 2.302, and 2.207) in our analysis failed to exceed the 8th corresponding mean eigenvalue (2.558, 2.481, 2.422, 2.373, 2.328, 2.287, 2.251, and 2.214) derived from 1,000 generated datasets with 438 cases and 168 variables, suggesting that a 7-factor solution was appropriate. This solution, along with item loadings, is detailed in S1 Table, using a maximum likelihood extraction and promax rotation given our expectation of correlated factors. We assessed meaningful loadings using criteria set at 0.32 ("poor"), 0.45 ("fair"), 0.55 ("good"), 0.63 ("very good"), and 0.71 ("excellent") [40]. Based on these criteria, a significant number of the 168 items loaded singularly above 0.32 on one of the factors, with several items loading above 0.32 on two or more factors, and only 2 items not loading significantly on any factor. The full analysis is presented in S1 Table.

The factor interpretation, conducted by the study's author, focused on the relationship between the psychological factors identified and their practical implications for sustainable farming. Each of the seven factors derived from the exploratory factor analysis offers a relatively unique view, being identified as separate factors within the EFA, on the motivations and barriers farmers face in adopting sustainable practices. The interpretation, which relied on the written content of items, helped align the theoretical frameworks with real-world applications. The seven factors from the Exploratory Factor Analysis can be described as follows. Factor 1 is a 'Sustainable Agricultural Commitment and Stewardship' factor. This factor encapsulates a multifaceted ethos of farmers' dedication to sustainability and their stewardship of the land. This factor includes statements reflecting personal satisfaction and alignment with values in achieving emission reduction targets, signifying a multifaceted approach to sustainable farming. It encompasses psychological motivations such as hedonic motivation, positive attitudes towards behavior, willingness, and a sense of competence in sustainable strategies, also highlighting community norms, personal growth, and autonomy in decision-making. Factor 2 is a 'Sustainable Farming Readiness and Confidence' factor. This factor encompasses statements about strategy integration and emission reduction targets, reflecting the farmers' readiness and capability in adapting to Net Zero practices. It draws on literacy about Net Zero, perceived behavioral control, and the creation of detailed strategies and coping mechanisms to address challenges. Factor 3 is a 'Sustainable Incentive Engagement and Acceptance' factor. This factor represents attitudes towards utilizing financial incentives for Net Zero practices. It

combines psychological constructs from the Unified Theory of Acceptance and Use of Technology, Theory of Planned Behavior, and Self-Determination Theory. Factor 4 is a 'Climate Adaptation Competence and Confidence in Agriculture' factor. Focusing on skills and confidence in climate adaptation and resilience strategies, this factor aligns with Net Zero Policy Commitment criteria and includes elements like competence, understanding, practicality, and coping strategies in climate adaptation. Factor 5 is a 'Net Zero Accountability and Reporting Commitment' factor. This factor focuses on adherence to Net Zero standards, especially in monitoring and reporting. It integrates psychological elements from several theories, reflecting a proactive orientation towards Net Zero practices. Factor 6 is a 'Community Influence and Commitment in Sustainable Farming' factor. Highlighting the role of social influence in complying with Net Zero frameworks and engaging with climate adaptation strategies, this factor emphasizes the impact of peer and community norms. Finally, Factor 7 is an 'Innovation and Technological Competence in Net Zero Farming' factor. This factor captures attitudes and abilities regarding the adoption and use of innovative technologies for Net Zero goals, aligning with concepts like self-efficacy and competence in utilizing new technologies.

### Short discussion

Study 1 developed an integrated psychological model to understand farmer attitudes towards six aspects of Net Zero policy commitments, integrating five psychological domains. Involving 438 UK farmers, the study utilized a comprehensive questionnaire with 168 items, each reflecting a facet of Net Zero policies and psychological factors. Exploratory Factor Analysis identified seven key factors: 'Sustainable Agricultural Commitment and Stewardship,' 'Sustainable Farming Readiness and Confidence,' 'Sustainable Incentive Engagement and Acceptance,' 'Climate Adaptation Competence and Confidence,' 'Net Zero Accountability and Reporting Commitment,' 'Community Influence and Commitment in Sustainable Farming,' and 'Innovation and Technological Competence.' This model, which we name the 'Integrated Motivation Model for Sustainable Farming,' offers new, integrated insights into the multifaceted psychological dimensions influencing farmers' approaches to Net Zero initiatives.

## Study 2

Study 2 aimed to build upon the Exploratory Factor Analysis (EFA) from Study 1. We developed a short measure the assess the Integrated Motivation Model for Sustainable Farming and conducted a Confirmatory Factor Analysis (CFA) to test the 7-factor model. Additionally, we examined the factor structure against indices of Net Zero Behavior to demonstrate the concurrent validity of the new measure.

### Method

**Sample.** The second study involved two samples:

1. 418 UK farmers: These respondents were a subset of Sample 1, recruited from Prolific using their Prolific ID. The gender distribution included 256 males, 161 females, and 1 individual who did not identify with either category. The average age of participants was 32.58 years, with a standard deviation of 12.06 years. Additional details of this sample can be found in Table 1.

2. 230 US farmers were recruited via Prolific following the same procedure as described for Study 1. The gender distribution comprised 137 males, 93 females, and 1 individual who did not identify with either category. The average age of participants was 32.75 years, with a standard deviation of 11.63 years. More information on this sample is provided in Table 1.

The recruitment period for participants was 6th– 11th March 2024. In alignment with ethical guidelines, informed consent was obtained from all participants. This consent was facilitated through an online survey, where respondents indicated their consent by affirmatively selecting the consent option on the digital form confirming consent statements. This consent option confirmed that participants were fully informed and voluntarily agreed to participate in the study.

**Measures.** To measure the seven dimensions identified in the Exploratory Factor Analysis (EFA), we developed a more concise 21-item version to assess the seven psychological domains identified in Study 1, which focus on psychological and behavioral change mechanisms. We named this the Integrated Motivation Model for Sustainable Farming Scale (items are listed in Table 3). The creation of a shorter scale was motivated by the need for efficient measurement. While a 168-item scale is comprehensive, it can be time-consuming and may lead to

**Table 3. Items for the Integrated Motivation Model for Sustainable Farming Scale.**

| Subscale | Items |
|---|---|
| Sustainable Agricultural Commitment and Stewardship | 1. "Being a farmer who helps achieve Net Zero is a core part of my identity."<br>2. "I feel proud when my farming practices contribute to Net Zero goals."<br>3. "For me, farming sustainably for Net Zero is about doing what I believe is right for the environment" |
| Sustainable Farming Readiness and Confidence | 4. "I feel ready and informed to change my farming methods to reach Net Zero."<br>5. "I am confident that I can adopt new farming practices that support Net Zero."<br>6. "Seeking knowledge to improve my farm's contribution to Net Zero is important to me." |
| Sustainable Incentive Engagement and Acceptance | 7. "Financial rewards for farming practices that support Net Zero align with my personal values."<br>8. "I believe using financial incentives is crucial for achieving Net Zero in my farming."<br>9. "Financial incentives are a key part of my strategy for farming towards Net Zero." |
| Climate Adaptation Competence and Confidence in Agriculture | 10. "I feel skilled in adapting my farming to address climate change and achieve Net Zero."<br>11. "Learning about climate adaptation strategies for Net Zero farming is a priority for me."<br>12. "I am confident in my ability to implement strategies that help my farm move towards Net Zero." |
| Net Zero Accountability and Reporting Commitment | 13. "I am committed to tracking and reporting my farm's progress towards Net Zero."<br>14. "I personally ensure that my farming practices are in line with Net Zero objectives."<br>15. "Honest and accurate reporting of my farm's activities is key to achieving Net Zero." |
| Community Influence and Commitment in Sustainable Farming | 16. "The farming practices of my community heavily influence my approach to achieving Net Zero."<br>17. "I feel a strong responsibility to my community to farm in ways that support Net Zero."<br>18. "Community support and ideas are essential for my commitment to Net Zero farming." |
| Innovation and Technological Competence in Net Zero Farming | 19. "I am confident in using new technologies to help my farm achieve Net Zero."<br>20. "Adopting innovative tools is crucial for my farm's journey towards Net Zero."<br>21. "I continuously upgrade my skills to effectively use technology for Net Zero farming." |

respondent fatigue, thus reducing the quality of responses. A shorter scale is more practical and user-friendly, increasing the likelihood of complete and thoughtful responses, thereby ensuring better data quality and reliability. In crafting a scale to measure seven key factors among farmers, we aimed to balance scientific robustness with practical usability, ensuring each factor is accurately represented without redundancy and respects the farmers' time and cognitive load. We opted for a concise 21-item scale, allotting three items per factor, as this is the minimum number of items required to ensure a factor is reliably measured [41]. This approach adheres to the standards of using Confirmatory Factor Analysis for psychometric analysis. Moreover, a shorter scale mitigates respondent fatigue and enhances participation and response quality and can be completed in no more than 5–10 minutes. This tailored, pragmatic design maintains scientific integrity and makes the scale an effective tool for practical application in varied farming contexts. The Integrated Motivation Model for Sustainable Farming Scale measures (i) Sustainable Agricultural Commitment and Stewardship, (ii) Sustainable Farming Readiness and Confidence, (iii) Sustainable Incentive Engagement and Acceptance, (iv) Climate Adaptation Competence and Confidence in Agriculture, (v) Net Zero Accountability and Reporting Commitment, (vi) Community Influence and Commitment in Sustainable Farming, and (vii) Innovation and Technological Competence in Net Zero Farming.

To assess the scale's concurrent validity, we aimed to correlate it with reports of actual behavior over the last 12 months. Specifically, we sought detailed behavior indices related to sustainable farming practices, rated on a scale from "0" to "7 or more," based on the frequency of occurrence. These indices were categorized under seven themes:

1. Sustainable Agricultural Commitment: "In the last year, how many new practices or technologies have you implemented on your farm that are specifically aimed at reducing emissions and promoting sustainability in terms of Net Zero?"

2. Sustainable Farming Readiness: "Over the past year, how many times have you attended workshops, courses, or training sessions related to Net Zero farming practices to improve your readiness and capability in adapting to sustainable agricultural methods?"

3. Sustainable Incentive Engagement: "In the last 12 months, how many different financial incentives or subsidy programs for Net Zero practices have you applied for or utilized on your farm?"

4. Climate Adaptation Competence: "How many distinct climate adaptation or resilience strategies have you actively integrated into your farming operations within the last year to combat the effects of climate change?"

5. Net Zero Accountability and Reporting: "Over the past year, how many times have you conducted formal assessments or reports of your farm's emissions in line with Net Zero accountability standards?"

6. Community Influence: "In the last year, how many collaborative projects or initiatives with other local farmers or community groups have you participated in to promote or enhance sustainable farming practices?"

7. Technological Innovation: "How many new technological solutions or innovations aimed at achieving Net Zero goals have you adopted or tested on your farm in the past 12 months?"

**Table 4. Fit statistics for the 7-factor the Integrated Motivation Model for Sustainable Farming Scale.**

|  | Chi-square | CMIN/DF | CFI | NNFI | RMSEA | SRMR |
|---|---|---|---|---|---|---|
| UK Sample | 358.40 | 2.133 | 0.965 | 0.956 | 0.052 | 0.038 |
| USA sample | 322.71 | 1.921 | 0.955 | 0.943 | 0.063 | 0.042 |

Key: CMIN/DF = Relative Chi-Square. CFI = Comparative Fit Index, NNFI = Non-normed Fit Index, RMSEA = Root Mean Square Error of Approximation, SRMR = Standardized Root Mean Square Residual

## Results

The sample size for both samples meets the criterion of 10 participants for every 1 item (e.g., both samples of n > 210), with a minimum sample size of $n = 100$ [42]. The Confirmatory Factor Analysis (CFA), using AMOS 28, was conducted using standard goodness-of-fit indices as recommended by [43, 44]. These indices include the relative chi-square (CMIN/DF), comparative fit index (CFI), non-normed fit index (NNFI), root mean square error of approximation (RMSEA), and standardized root mean square residual (SRMR). We assessed the model's fit by determining if the fit statistics represented an acceptable fit based on established criteria: a CMIN/DF of less than 3, CFI and NNFI greater than 0.90, and RMSEA and SRMR less than 0.08 [43, 45, 46]. The findings for both models are presented in Table 4. These values indicate that the model suggests a good fit to the data, meeting the criteria, validating the factor structure identified in the Exploratory Factor Analysis (EFA), and supporting the structural validity of the 21-item Net Zero Farming Alignment Assessment.

Table 5 shows the relationship between scores on the Integrated Motivation Model for Sustainable Farming Scale and indicators of actual behaviors in the context of net-zero farming. We report statistical significance using a frame of reference, with $r \geq .37$ representing a large effect size, $.24 \leq r < .37$ representing a moderate effect size, and $.1 \leq r < .24$ representing a small effect size [47, 48]. A moderate effect size is deemed to be the minimum at which the findings can be considered of practical significance [49]. This criterion differs from the well-

**Table 5. Pearson product-moment correlations between the 7-factor Integrated Motivation Model for Sustainable Farming Scale and reports of Net-Zero activities within the last year.**

|  | Practices | Readiness | Incentive Engagement | Climate Adaptation | Accountability and Reporting | Community Influence | Technological Innovation |
|---|---|---|---|---|---|---|---|
| 1. Sustainable Agricultural Commitment and Stewardship | .390** | .288** | .353** | .398** | .359** | .324** | .391** |
| 2. Sustainable Farming Readiness and Confidence | .334** | .292** | .328** | .334** | .358** | .304** | .403** |
| 3. Sustainable Incentive Engagement and Acceptance | .408** | .316** | .309** | .329** | .345** | .275** | .352** |
| 4. Climate Adaptation Competence and Confidence in Agriculture | .459** | .382** | .404** | .394** | .404** | .348** | .461** |
| 5. Net Zero Accountability and Reporting Commitment | .461** | .373** | .413** | .390** | .449** | .327** | .449** |
| 6. Community Influence and Commitment in Sustainable Farming | .400** | .356** | .400** | .361** | .412** | .389** | .431** |
| 7. Innovation and Technological Competence in Net Zero Farming | .439** | .295** | .342** | .342** | .326** | .258** | .397** |

Key
* p < .05
** p < .01

cited effect size of .1 = small to .5 = large. Cohen based the comparisons with the d effect size criteria using a biserial correlation. Comparison with the d effect size for Pearson product moment correlation coefficients should be based on point biserial correlation [48].

By assessing the correlations across seven factors with reported behavior, the findings reveal a consistent pattern: there is a positive correlation across all domains. This indicates that an increase in the score within these domains is associated with a higher likelihood of engaging in various sustainable behaviors or initiatives. The strength of these correlations varied, mostly falling within the medium to large effect size range, which underscores the substantive relationships between the attitudes and behaviors and the reported practices. In terms of the Sustainable Agricultural Commitment and Stewardship subscale, the findings suggest moderate correlations ranging from $r = .324$ for community involvement to $r = .390$ for the adoption of new practices or technologies. This suggests that farmers committed to sustainable agriculture are moderately likely to engage in both collaborative efforts and the adoption of new practices. For the Sustainable Farming Readiness and Confidence subscale, the correlations indicate moderate relationships, the highest being $r = .403$ for technological solutions or innovations and the lowest being $r = .304$ for collaborative projects. It's clear that farmers who feel ready and confident are more inclined to adopt technological innovations and collaborate on projects aimed at sustainability. For the Sustainable Incentive Engagement and Acceptance subscale, a notable finding is the large correlation ($r = .408$) with incentives or subsidy programs, suggesting a strong relationship between farmers' engagement with sustainable incentives and their behaviors. This indicates that incentives or subsidy programs are effective motivators for adopting sustainable practices. For Climate Adaptation Competence and Confidence in Agriculture, this factor shows predominantly moderate to large effect sizes in its correlations, ranging from $r = .348$ for formal assessments to $r = .461$ for technological solutions or innovations. It implies that farmers who are competent and confident in their ability to adapt to climate change are significantly more likely to engage in a range of net-positive behaviors. For Net Zero Accountability and Reporting Commitment, the correlations here are consistently moderate to large, with a peak at $r = .449$ for behaviors such as formal assessments and the adoption of technological solutions or innovations. This strong concurrent validity suggests that a commitment to accountability and reporting is closely linked to sustainable actions. For the Community Influence and Commitment in Sustainable Farming subscale, the correlations demonstrate moderate to large correlations, with the largest being $r = .431$ for technological solutions or innovations. This factor highlights the significant role of community influence and commitment in promoting sustainable farming practices. Finally, for the Innovation and Technological Competence in Net Zero Farming subscale, this shows a moderate to large correlation with attended workshops ($r = .295$) and a significant correlation with technological solutions or innovations ($r = .397$). It emphasizes the importance of innovation and technological competence in driving sustainable practices and the adoption of net-zero initiatives.

Finally, an independent groups t-test was conducted to compare the scores on each of the scales between farmers in the USA and the UK. The decision to use an independent t-test was supported by the skewness statistics for each group, which ranged from -0.85 to -0.358, indicating a normal distribution. Table 6 presents the mean scores, standard deviations, t-values, p-values, and effect sizes for these tests. Considering multiple comparisons were made across seven psychological scales, a Bonferroni correction was applied to adjust the significance level to p < 0.007. Using this stringent criterion, significant differences in the scores for 'Sustainable Farming Readiness and Confidence' and 'Sustainable Incentive Engagement and Acceptance' were found. In assessing the magnitude of these differences between USA and UK farmers, effect sizes were evaluated using Cohen's d (49), with values of 0.2, 0.5, and 0.8 representing small, medium, and large effects, respectively." In terms of the effect sizes of these significant

**Table 6. Mean (Standard deviation) comparisons between UK and USA farmers for the Integrated Motivation Model for Sustainable Farming Subscales.**

| | UK (n = 418) | | USA (n = 230) | | | | |
|---|---|---|---|---|---|---|---|
| | Mean | SD | Mean | SD | t | p | d |
| Sustainable Agricultural Commitment and Stewardship | 11.47 | 2.11 | 11.18 | 2.46 | 1.57 | 0.117 | 0.13 |
| Sustainable Farming Readiness and Confidence | 9.70 | 2.36 | 10.60 | 2.48 | -4.59 | 0.001* | 0.37 |
| Sustainable Incentive Engagement and Acceptance | 8.97 | 2.41 | 11.11 | 2.44 | -10.78 | 0.001* | 0.89 |
| Climate Adaptation Competence and Confidence in Agriculture | 10.72 | 2.29 | 10.65 | 2.55 | 0.37 | 0.711 | 0.03 |
| Net Zero Accountability and Reporting Commitment | 10.58 | 2.34 | 10.97 | 2.52 | -1.93 | 0.054 | 0.16 |
| Community Influence and Commitment in Sustainable Farming | 10.12 | 2.46 | 10.55 | 2.56 | -2.10 | 0.036 | 0.17 |
| Innovation and Technological Competence in Net Zero Farming | 10.61 | 2.33 | 10.99 | 2.48 | -1.93 | 0.055 | 0.16 |

* p < .007 (Significance level with Bonferroni correction due to 7 comparisons).

differences, the effect size for 'Sustainable Farming Readiness and Confidence' was small, and the effect size for 'Sustainable Incentive Engagement and Acceptance' was medium.

## Short discussion

The findings from the Confirmatory Factor Analysis (CFA) confirmed the structural validity of the 7-factor Integrated Motivation Model for Sustainable Farming Scale, meeting standard goodness-of-fit indices across UK and USA samples. The correlations between the 7 factors of the Integrated Motivation Model for Sustainable Farming Scale and reports of NetZero behavioral actions within the last year generally showed medium to large effect sizes, suggesting the concurrent validity of the scale.

## Discussion

Formulating a seven-factor model of psychological approaches to sustainable farming among farmers, that we name the "Integrated Motivation Model for Sustainable Farming", represents a breakthrough in deciphering the psychological intricacies and pragmatic dimensions of sustainable agriculture under a Net Zero framework. The seven factors are:

1. Sustainable Agricultural Commitment and Stewardship. This factor delves into farmers' personal satisfaction and their alignment with values pertinent to achieving emission reduction targets. This factor encapsulates a broad spectrum of psychological theory, including hedonic motivation, positive attitudes towards sustainable behavior, willingness, and competence [1, 3, 10]. It also emphasizes the importance of community norms, personal growth, and autonomy in decision-making [3, 18, 19]. Therefore, this factor reflects a comprehensive dimension in motivational, cognitive, and social processes towards sustainable farming.

2. Sustainable Farming Readiness and Confidence. This factor addresses the integration of strategies and the attainment of emission reduction targets, spotlighting the farmers' readiness and capability in adapting to Net Zero practices. This factor includes an emphasis on Net Zero literacy, perceived behavioral control, and the development of detailed strategies and coping mechanisms to navigate potential challenges [3, 11, 24].

3. Sustainable Incentive Engagement and Acceptance. This factor revolves around the attitudes of farmers towards financial incentives linked to Net Zero practices, blending insights from the Unified Theory of Acceptance and Use of Technology [1], the Theory of Planned Behavior [3], and Self-Determination Theory [7].

4. Climate Adaptation Competence and Confidence in Agriculture. This factor concentrates on the skills and confidence necessary for climate adaptation and resilience strategies, aligning with Net Zero Policy Commitment criteria. This factor brings in elements like competence, understanding, practicality, and the ability to formulate effective coping strategies for climate adaptation [3, 11].

5. Net Zero Accountability and Reporting Commitment. This factor focuses on adherence to Net Zero standards, particularly in the realms of monitoring and reporting, integrating various psychological theories [1, 3, 7, 11, 19] to reflect a proactive approach towards Net Zero practices.

6. Community Influence and Commitment in Sustainable Farming. This factor underscores the significant role of social influence and community norms [3, 7] in adhering to Net Zero frameworks and implementing climate adaptation strategies.

7. Innovation and Technological Competence in Net Zero Farming. This factor captures the attitudes and capabilities of farmers in adopting and utilizing innovative technologies to meet Net Zero goals, highlighting key concepts like self-efficacy and competence [1, 7, 19].

We introduce the Integrated Motivation Model for Sustainable Farming, which defines seven key factors by merging two distinct frameworks presented in this paper: Behavioral-Adoption and Purpose-Driven. The Behavioral-Adoption framework is focused on enhancing practical readiness, skill development, and strategic implementation necessary for adopting sustainable practices. It emphasizes external motivations such as incentives, compliance with standards, and new technology utilization. The five factors identified in the study—Sustainable Farming Readiness and Confidence, Sustainable Incentive Engagement and Acceptance, Climate Adaptation Competence and Confidence in Agriculture, Net Zero Accountability and Reporting Commitment, and Innovation and Technological Competence in Net Zero Farming—represent this framework. They underscore the importance of equipping farmers with the necessary tools and information, creating environments conducive to sustainable practices, and ensuring farmers feel competent and supported. Conversely, the Purpose-Driven framework delves into intrinsic and value-based motivations driving sustainable farming. It aligns farmers' actions with their personal values and community norms, fostering deeper engagement and commitment to sustainability. Factors such as Sustainable Agricultural Commitment and Stewardship, and Community Influence and Commitment in Sustainable Farming reflect this framework, focusing on internal satisfaction and personal fulfillment driven by a responsibility towards the community and environment. The interplay of these frameworks within our model offers a dual perspective that captures the motivational, cognitive, and social processes influencing sustainable farming. This integration helps us understand not only the practical steps necessary for behavioral adoption but also the deeper sense of purpose sustaining these changes over time. By illuminating the complex interplay between Behavioral-Adoption and Purpose-Driven factors, our model provides a holistic view of what drives farmers towards achieving Net Zero goals. It becomes a crucial tool for researchers, policymakers, and practitioners in the field of sustainable agriculture.

The formulation of the Integrated Motivation Model for Sustainable Farming into a novel 21-item scale (the Integrated Motivation Model for Sustainable Farming Scale), underpinned by a comprehensive model of psychological motivations, cognitions, and social processes, marks a significant advancement in quantifying seven critical psychological dimensions of farmers' engagement with Net Zero practices. The structural validity of this scale was confirmed through a rigorous process of confirmatory factor analysis across two samples, solidifying its efficacy in assessing the complex interplay between psychological factors and

sustainable farming practices in pursuit of Net Zero targets. Study 2 also demonstrates a positive correlation among the seven factors of the Agricultural Net Zero Adaptation and Resilience Scale related to reports of sustainable farming practices over the past year, indicating that higher scores correlate with a greater likelihood of engaging in sustainable behaviors. The correlations ranged from medium to large, highlighting significant relationships among attitudes, readiness, and actions in sustainability, including engagement with incentives and technological innovations. This underscores that fostering commitment, readiness, engagement, competence, and confidence in sustainable and adaptive agricultural practices is related to behaviors that align with sustainability and climate adaptation goals. The significant associations between psychological factors and Net Zero farming practices underscore the potential for the scale to inform targeted interventions. For example, enhancing farmers' confidence and readiness through tailored educational programs could facilitate the adoption of sustainable practices. These findings highlight the practical implications for developing support mechanisms that align with farmers' profiles as assessed by the Integrated Motivation Model for Sustainable Farming.

The findings from employing both UK and USA samples revealed significant differences in two of the subscales of the Integrated Motivation Model for Sustainable Farming scale: 'Sustainable Farming Readiness and Confidence' and 'Sustainable Incentive Engagement and Acceptance,' with farmers in the USA scoring significantly higher and demonstrating small and medium effect sizes, respectively. The 'Sustainable Farming Readiness and Confidence' factor encompasses aspects such as strategy integration, emission reduction targets, and a comprehensive understanding of Net Zero practices, reflecting farmers' readiness and capabilities in adapting to these practices. The higher scores among farmers in the USA could be attributed to well-established agricultural extension services in the United States [50], which may better equip farmers with the necessary literacy about Net Zero practices and the skills to implement detailed strategies and coping mechanisms. Similarly, the 'Sustainable Incentive Engagement and Acceptance' factor, which assesses attitudes toward utilizing financial incentives for Net Zero practices, also showed higher scores among farmers in the USA. This may be explained by the differences in agricultural policy frameworks between the two countries. In the United States, there is a robust system of financial incentives for sustainable farming under various federal and state programs, likely reflected in the higher engagement and acceptance scores. Meanwhile, the agricultural policy framework in the UK has been undergoing significant changes with new schemes currently being introduced following the Agriculture Act 2020 [51] and the exit from the European Union's Common Agricultural Policy.

This study's innovative approach, which maps psychological criteria onto farmers' Net Zero commitments, represents a significant advancement in agricultural climate policy. It integrates diverse criteria by combining elements of Net Zero Policy Commitment with well-cited psychological theories, including the Unified Theory of Acceptance and Use of Technology, the Theory of Planned Behavior, the Prototype-Willingness Model, Implementation Intentions, Literacy, Eudaimonia, and Self-Determination Theory. This holistic view goes beyond the traditional focus areas of policy and technology, delving into the psychological and behavioral aspects of farmers. The study covers a comprehensive range of Net Zero commitment areas, from emission reduction targets to financial incentives and adaptation strategies, ensuring a deep understanding of farmers' perceptions and interactions with these aspects. Therefore, this model offers a nuanced yet parsimonious understanding of the intricate relationship between policy requirements and the psychological dimensions of farmers' behaviors and attitudes towards Net Zero commitments. This approach could lead to more effective strategies for implementing these commitments in the agricultural sector by considering how the practical and psychological aspects of sustainable farming transitions align. To illustrate the practical

applications of our findings, the Integrated Motivation Model for Sustainable Farming scale could be utilized by policymakers to design interventions that closely align with farmers' psychological needs. For example, if the Integrated Motivation Model for Sustainable Farming reveals a strong influence of community norms on farmer behavior, policymakers could initiate community-led programs that leverage peer influences to promote sustainable practices. These programs might include peer mentoring systems where experienced farmers who have successfully adopted sustainable techniques share their practices and experiences. Such community-focused interventions can harness the collective power, enhancing the acceptance and integration of sustainable practices within local farming communities.

However, it is important to acknowledge the limitations of the study, including: 1) its broad nature, 2) its cross-sectional design, and 3) its focus on psychological mechanisms. First, given the broad scope of the study, exploring the role of regional farming and policy environments would provide valuable context to our findings. Examining the impact of regional policy environments in a more granular manner could reveal how specific policies influence psychological outcomes and sustainable behaviors in diverse farming contexts. Future studies could then offer more customized and effective recommendations for policy adjustments and support mechanisms tailored to regional needs. Second, regarding the cross-sectional design, future research could employ longitudinal designs to track the evolution of farmers' attitudes and behaviors over time. This approach would allow researchers to better ascertain causality and the durability of psychological factors influencing sustainable farming practices. Third, concerning the focus on psychological mechanisms, although the work has established integrated models across various psychological contexts, advocating for interdisciplinary collaboration is crucial to further enhance our understanding of sustainable agriculture. The integration of insights from psychology, agronomy, economics, and environmental science not only enriches our approach but is essential in tackling the multifaceted challenges associated with achieving Net Zero emissions in agriculture. For instance, while agronomic innovations provide the tools for sustainable farming, psychological insights into farmer behavior and economic analyses of market and policy impacts are equally critical for the adoption and effectiveness of these technologies. Environmental science connects these elements to the broader ecological outcomes, ensuring that sustainable agricultural practices align with global environmental goals. Together, integrating these disciplines contributes to a comprehensive strategy that addresses both the practical and theoretical aspects of sustainability, creating a resilient agricultural framework that can adapt and thrive in the face of climate challenges. To some extent, the current findings demonstrate the potential for adopting this interdisciplinary approach.

In summary, the current study has unveiled a comprehensive understanding that intersects psychological underpinnings with agricultural practices, aiming at Net Zero targets. The study's novel integration of psychological theories offers an enriched perspective on farmers' engagement with sustainable practices, underlining the complex interplay between individual motivations, community norms, and technological competencies through Behavioral-Adoption and Purpose-Driven motivations. The formulation and measurement of the Integrated Motivation Model for Sustainable Farming, rooted in empirical evidence, illuminates the multifaceted psychological dimensions influencing farmers' sustainable behaviors. Our research underscores the crucial role of psychological factors in driving the agricultural sector toward Net Zero, suggesting that future policies and interventions must consider these insights to foster a more sustainable and resilient agricultural landscape. This approach not only potentially enhances the effectiveness of sustainability initiatives but also charts a path for a unified strategy that bridges the gap between psychological insights and practical agricultural applications, offering a roadmap for achieving sustainability in farming practices. The findings underscore the potential for targeted interventions that leverage psychological insights to foster sustainable

farming practices. By understanding the motivational and cognitive processes that influence farmers' behavior, policymakers and practitioners can design more effective strategies that not only promote environmental sustainability but also support the well-being of farmers.

## Supporting information

**S1 Table. Exploratory factor analysis using maximum likelihood extraction and promax rotation of the NetZero farming items.**
(DOCX)

## Author Contributions

**Conceptualization:** John Maltby.

**Data curation:** John Maltby.

**Formal analysis:** John Maltby.

**Funding acquisition:** John Maltby.

**Investigation:** John Maltby.

**Methodology:** John Maltby.

**Project administration:** John Maltby.

**Writing – original draft:** John Maltby.

**Writing – review & editing:** John Maltby.

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
