## [Decision Letter · Decision Letter 0]

16 Apr 2024

PONE-D-24-11759Avoiding Silo Thinking: Integrating Psychological Insights into Farming Net Zero Policy Commitments.PLOS ONE

Dear Dr. Maltby,

Thank you for submitting your manuscript to PLOS ONE. After careful consideration, we feel that it has merit but does not fully meet PLOS ONE’s publication criteria as it currently stands. Therefore, we invite you to submit a revised version of the manuscript that addresses the points raised during the review process.

We look forward to receiving your revised manuscript.

Kind regards,

Amar Razzaq, PhD

Academic Editor

PLOS ONE

Journal Requirements:

Additional Editor Comments (if provided):

Please improve the abstract, Introduction and Disucssion section. Abstract is very technical. It should reflect the data sources, methods, and the main findings of the study as well the policy implications. Furthermore, clarify what is silo thinking and improve upon it in the Introduction section while accentuating the research gap and relevant literature. State the obejctives clearly. For improvement in Discussion section, see reviewers' comments.

Reviewers' comments:

Reviewer's Responses to Questions

**Comments to the Author**

1. Is the manuscript technically sound, and do the data support the conclusions?

Reviewer #1: Yes

Reviewer #2: Yes

2. Has the statistical analysis been performed appropriately and rigorously? 

Reviewer #1: Yes

Reviewer #2: Yes

3. Have the authors made all data underlying the findings in their manuscript fully available?

Reviewer #1: Yes

Reviewer #2: Yes

4. Is the manuscript presented in an intelligible fashion and written in standard English?

Reviewer #1: Yes

Reviewer #2: Yes

5. Review Comments to the Author

Reviewer #1: written neatly and justified the title. however, the author can improve over statistical approach to meet the objectives of the study. The introduction is well-written and provides a good overview of the topic. However, the second paragraph is a bit too long and could be shortened by combining some of the sentences. Additionally, the third paragraph could be improved by adding a few more specific examples. The conclusion is well-written and summarizes the main points of the paper well. However, I would suggest adding a sentence or two about the implications of the findings.

Reviewer #2: Avoiding Silo Thinking: Integrating Psychological Insights into Farming Net Zero Policy Commitments

Abstract: The study presented offers a comprehensive examination of the psychological drivers influencing farmers' adoption of Net Zero policies, incorporating a wide range of theoretical frameworks. This multi-dimensional approach, encompassing theories such as the Unified Theory of Acceptance and Use of Technology and Self-Determination Theory, enriches our understanding of farmers' attitudes and behaviors towards sustainability initiatives. However, while the integration of diverse psychological perspectives is commendable, it may be beneficial to provide a clearer rationale for the selection of specific theories and how they interact within the context of sustainable agriculture. Additionally, the findings from Study 1, elucidating seven latent factors influencing farmers' engagement with Net Zero policies, provide valuable insights into the nuanced motivations and challenges faced by agricultural practitioners. Nevertheless, further elaboration on the operationalization of these factors and their practical implications for policy development and intervention strategies would enhance the study's applicability and relevance to stakeholders in the agricultural sector. Moreover, Study 2's validation of the proposed psychological model through Confirmatory Factor Analysis and concurrent validity assessment represents a significant contribution to the field. However, considering the geographical variation in agricultural practices, particularly between the UK and the US, it would be informative to explore potential cross-cultural differences in the psychological determinants of Net Zero farming. Overall, this research offers a robust framework for understanding the complex interplay of psychological factors shaping farmers' adoption of sustainable practices, laying the groundwork for future investigations and practical interventions aimed at promoting environmental stewardship in agriculture.

Introduction: The passage effectively integrates a wide array of psychological frameworks to elucidate the complexities of farmers' adoption of sustainable practices. However, to enhance clarity and coherence, it could benefit from a more structured presentation of these frameworks, perhaps organized into distinct sections or subsections based on thematic similarities or theoretical foundations. This structural refinement would assist readers in navigating the diverse theoretical perspectives and their interconnections more seamlessly.

Furthermore, while the passage highlights the importance of simplifying and unifying psychological frameworks for practical application, it could provide more concrete examples or case studies illustrating how this integration process could be operationalized in real-world contexts. By demonstrating the application of the proposed unified model in specific agricultural settings or initiatives, readers would gain a clearer understanding of its potential implications and utility for stakeholders in the field.

Additionally, the passage could delve deeper into the potential challenges or limitations associated with integrating multiple psychological frameworks, such as conflicting theoretical assumptions or methodological differences. Acknowledging and addressing these challenges would add depth to the discussion and provide a more nuanced understanding of the complexities involved in developing a comprehensive psychological model for sustainable agriculture.

Finally, while the passage outlines the methodological approach for developing and validating the proposed psychological model, it could provide more detail on the specific steps involved in data collection, analysis, and interpretation. Providing transparency regarding the research methodology would enhance the credibility and replicability of the study, enabling readers to assess the rigor of the findings and their applicability to other contexts.

The discussion effectively outlines the seven-factor model and the development of the ANZAR Scale, providing valuable insights into the psychological dimensions of farmers' engagement with Net Zero practices. However, to enhance clarity and coherence, it could benefit from a more structured presentation, perhaps organizing the discussion around each factor or scale development process individually. This would help readers navigate the complex interplay of psychological theories and empirical findings more seamlessly, facilitating a deeper understanding of the research outcomes.

Additionally, while the discussion highlights the practical implications of the findings for policymakers and practitioners, it could provide more specific examples or case studies illustrating how the ANZAR Scale could be utilized in real-world contexts. Concrete examples demonstrating how the scale could inform the design of targeted interventions or support mechanisms tailored to farmers' psychological profiles would enhance the applicability of the research findings and offer actionable insights for stakeholders in the agricultural sector.

Furthermore, the discussion acknowledges the limitations of the study, such as its cross-sectional nature and the need for future research to explore longitudinal designs and regional policy environments. While these limitations are appropriately addressed, providing suggestions for how future studies could mitigate or overcome these limitations would add depth to the discussion and guide researchers in designing more robust investigations in the future.

Finally, the discussion could emphasize the importance of interdisciplinary collaboration in advancing research on sustainable agriculture. Highlighting the potential benefits of integrating insights from disciplines such as psychology, agronomy, economics, and environmental science could underscore the need for a holistic approach to addressing the complex challenges of achieving Net Zero emissions in agriculture.

In summary, while the discussion provides a thorough examination of the research findings and their implications, further improvements could be made to enhance the clarity, applicability, and interdisciplinary perspective of the discussion. By incorporating these suggestions, the discussion could offer a more comprehensive and impactful contribution to the field of sustainable agriculture.

6. PLOS authors have the option to publish the peer review history of their article (what does this mean?). If published, this will include your full peer review and any attached files.

Reviewer #1: **Yes: **Dr. navghan Mahida

Reviewer #2: No

---

## [Author Response · Author response to Decision Letter 0]

3 May 2024

RESPONSE TO REVIEWERS

Thank you for considering the recent submission and for your encouraging response regarding the revision and resubmission of the manuscript. I extend my gratitude to the reviewers for their insightful, helpful, patient, and constructive feedback. In response to the reviewers' comments, I have thoroughly reviewed the paper and made numerous significant changes and additions. I have also corrected some typing errors and I apologise for the presence of those. In also revising the paper, and making corrections, I have provided the exact dates for data collection as submission guidelines and amended the name of the scale (to the Integrated Motivation Model for Sustainable Farming scale), to make it more concordant with the revisions. To provide clarity the revised text corresponding to each response is included alongside the response with the location of the change in the revised manuscript also presented.

Additional Editor Comments 

1. Please improve the abstract, Introduction and Disucssion section. Abstract is very technical. It should reflect the data sources, methods, and the main findings of the study as well the policy implications. Furthermore, clarify what is silo thinking and improve upon it in the Introduction section while accentuating the research gap and relevant literature. State the obejctives clearly. For improvement in Discussion section, see reviewers' comments.

>>>>> We have rewritten the abstract complete and it reads on Page 2, Line 2 to Page 3. Line 2 of the revised manuscript as follows:

“This study enhances our understanding of the psychological factors influencing farmers' adoption of sustainable farming practices, specifically those related to achieving NetZero emissions. It achieves this by integrating various psychological theories with practical farming methods within the context of Behavioral-Adoption and Purpose-Driven contexts. The research consisted of two studies. Study 1 employed Exploratory Factor Analysis (EFA) to analyze responses from 438 UK farmers regarding their attitudes toward a series of Net Zero policy commitments, drawing on psychological theories including the Unified Theory of Acceptance and Use of Technology, the Theory of Planned Behavior, the Prototype Willingness Model, Implementation Intentions, Self-Determination Theory, Eudaimonia, and the Integrated Model of Health Literacy. The findings revealed a new Integrated Motivation Model for Sustainable Farming that comprises seven factors: Agricultural Commitment and Stewardship, Sustainable Farming Readiness and Confidence, Sustainable Incentive Engagement and Acceptance, Climate Adaptation Competence and Confidence, Net Zero Accountability and Reporting Commitment, Community Influence and Commitment in Sustainable Farming, and Innovation and Technological Competence. Study 2 validated these factors through the development of a 21-item Integrated Motivation Model for Sustainable Farming scale and use of Confirmatory Factor Analysis (CFA) to confirm the 7-factor structure using a subsample of 418 UK farmers from Study 1 and an additional 230 US farmers. Furthermore, Study 2 tested the concurrent validity of the new scale by demonstrating significant associations with reported sustainable farming behaviors. These findings underscore the complex interplay of motivational, cognitive, and social processes influencing sustainable farming practices. The integrated psychological model developed through this research provides parsimonious and valuable insights for policymakers to foster sustainable practices in farming effectively. The confirmation of this model across farming populations enhances its generalizability and potential to guide targeted interventions aimed at achieving behavioral change in pursuit of Net Zero targets in agriculture.”

2. Furthermore, clarify what is silo thinking and improve upon it in the Introduction section while accentuating the research gap and relevant literature. 

>>>>> In response to the reviewer's suggestion, we have expanded the discussion of 'silo thinking' – though we’ve re-termed it has “silo approaches” in the Introduction to clarify its meaning and implications within the context of sustainable agriculture. This revision appears on Page 6, Lines 18 to 21 of the revised manuscript. 

“By identifying common themes across these frameworks, we can simplify complex theories into practical strategies that prevent siloed approaches (an approach where different sectors or disciplines operate in isolation) and enhance farmers' and land managers' adoption of sustainable practices.”

We have also enhanced the section by explicitly linking the concept to the identified research gap and discussing relevant literature, a revision also largely shaped by the first comment from Reviewer 2. This amendment not only addresses the reviewer's concerns but also strengthens the manuscript by providing a clearer exposition of the theoretical underpinnings of our research. It demonstrates the necessity and impact of a multidisciplinary approach in addressing complex agricultural challenges, incorporating Behavioral-Adoption and Purpose-Driven frameworks. We have added this aspect to the paragraph. This revision appears from Page 6, Line 21 to Page 7, Line 14 of the revised manuscript. 

“This integration helps bridge the gap between intricate psychological theories and farming realities, promoting a more cohesive understanding. For example, in regions where farming demands significant water resources, leading to environmental stress, especially during drought conditions, promoting water-efficient technologies poses a challenge (21). Farmers may be hesitant due to high upfront costs, uncertainty about effectiveness, and disruption of established practices. A better understanding can be achieved by integrating factors from Behavioral-Adoption and Purpose-Driven frameworks. We could identify factors that emphasize practical readiness and external factors such as financial incentives (Behavioral-Adoption), which align with farmers' values towards sustainability and stewardship of natural resources (Purpose-Driven). This integration provides a more meaningful and strategic understanding of these factors. Similarly, in areas where farming is crucial to the economy and vulnerable to climate change (22), a unified model that integrates Behavioral-Adoption strategies—demonstrating the effectiveness and efficiency of new sustainable practices—with the Purpose-Driven aspects that engage farmers' values and cultural practices related to land and community welfare, can be advantageous. This dual approach ensures farmers are not only prepared but also motivated to adopt new practices that benefit both their livelihood and their community. Thus, this unified approach addresses both the logistical and motivational aspects of adopting or changing behaviors to target sustainable farming.”

3. State the objectives clearly. 

>>>>> In response to the reviewer’s comment, we have clarified the objectives of our study by explicitly listing them and detailing the methodological approaches used to achieve each goal. The revision includes a structured breakdown of the study's aims, enhancing the clarity and readability of our research intentions. This change addresses the need for a precise and comprehensible presentation of our study’s goals, ensuring that the objectives are unmistakably clear and logically presented. This amendment will aid in better understanding the scope and direction of our research within the broader context of psychological studies in sustainable agriculture. This revision appears on Page 8, Line 11 to Page 9, Line 5 of the revised manuscript.

“The primary aim of this study is to develop a comprehensive psychological model that merges the Behavioral-Adoption and Purpose-Driven frameworks around sustainable farming practices to achieve NetZero among farmers. This integrated model will enhance our understanding of the psychological factors that influence farmers’ behaviors and motivations towards sustainable farming practices to achieve NetZero goals. Specifically, there are three objectives:

1. Develop an integrated psychological framework that integrates diverse perspectives to explain behavior change in the context of sustainable farming. This framework will utilize exploratory factor analysis (EFA) as an abductive reasoning method (26) to identify underlying psychological factors from a broad array of existing psychological models and constructs that reflect Behavioral-Adoption and Purpose-Driven frameworks.

2. Demonstrate structural validity for the proposed integrated psychological model through confirmatory factor analysis (CFA) (27), , ensuring that our model reliably captures the essential motivational, cognitive, and social processes that influence farmers’ decisions.

3. Assess the model’s practical implications by linking the psychological factors to actual behavioral indicators of NetZero farming practices, thereby demonstrating the model’s concurrent validity.”

4. For improvement in Discussion section, see reviewers' comments.

>>>>> "In our revised manuscript, we have taken on board the reviewers' comments, which are detailed below. We have elaborated on the operationalization of factors affecting net zero emissions in agriculture, providing detailed practical implications for policy development and intervention strategies. We structured the discussion around individual factors and the scale development process to enhance understanding of the interplay between psychological theories and empirical findings. Our study addresses its limitations and offers suggestions for future research to build more robust methodologies. Additionally, we suggest that this provides confidence for future work looking towards integrated insights from psychology, agronomy, economics, and environmental science, advocating for a holistic approach to tackle the complex challenges of achieving net zero emissions in the agricultural sector.

Reviewer #1:

1. However, the second paragraph is a bit too long and could be shortened by combining some of the sentences. 

>>>>> Thank you for the suggestion. Due to other comments, we have rewritten the introduction, so the structure and content are different – in which we make a case for grouping psychological theories into behavioural adoption and purpose-drive frameworks. Nonetheless, we have been mindful about the length of paragraphs. 

2. Additionally, the third paragraph could be improved by adding a few more specific examples. 

>>>>> In the revised manuscript, we have structured the work into Behavioural-adoption and Purpose-driven theories. This paragraph – which is now the fourth - is now part of the purpose-driven theories. However, I have expanded this paragraph by adding more specific examples of how theories like the Theory of Implementation Intentions can be practically applied, particularly in agricultural contexts. Examples include the successful implementation of detailed action plans for sustainable practices like pesticide reduction and water conservation in diverse geographical areas. This addition aims to demonstrate the real-world applicability and effectiveness of the integrated theoretical models discussed, thereby addressing your valuable suggestion to enhance the specificity of examples provided in the manuscript. This revised paragraph appears on Page 5, Line 11 to Page 6, Line 13 of the revised manuscript as follows: 

“The second set to consider are Purpose-Driven frameworks that focus on beliefs, meaning, fulfillment, desired prototypes, environmental literacy processes, and contributions to personal and community well-being. Within the existing sustainable farming literature, Self-Determination Theory (SDT;(7)) has been used to explore practices in sustainable farming as reflections of purpose-driven intentions in farming, in terms of motivational intrinsic versus extrinsic needs driven by autonomy, competence, and relatedness. Rsearch shows that SDT principles are instrumental in how a self-directed approach contributes to sustainable farming, emphasizing how intrinsic choices over external rewards and fostering conditions that support farmer autonomy leads to sustainable farming (8,9). Moreover, integrating models such as the Prototype Willingness Model (10) and the theory of Implementation Intentions (11,12) could offer an advanced understanding of concepts of prototypical influences around the identity of being a sustainable farmer (e.g., the ‘eco-friendly farmer’) and intentional, planned behaviors (e.g. specific plans) in sustainable farming are Purpose-Driven factors in farming. For example, in terms of implementation intentions, developing detailed action plans for pesticide reduction or water conservation can underline the practical applicability of these theories. (13,14). There are also other psychological framework that we consider Purpose-Driven. The inclusion of models of environmental literacy (15) that enriches our understanding by framing the competencies necessary for sustainable farming. This approach not only emphasises individuals' abilities to access and understand information relating to sustainable farming, but also emphasis the ability to make purposeful and well-informed decisions that positively impact on sustainable farming. Lastly, though there has been an exploration of the relationship between everyday affect (e.g. enjoyment) and farming practices (16,17) there is a compelling need for a broader examination of well-being concepts, particularly eudaimonia. Eudaimonia focuses on fulfilling one's potential and finding deep meaning and happiness in alignment with one’s true self (18,19). Exploring how eudaimonia correlates with sustainable farming practices (20) offers vital insights into possibly why fulfilment and meaning may be purposefully connected with the adoption of sustainable practices in farming.”

3. The conclusion is well-written and summarizes the main points of the paper well. However, I would suggest adding a sentence or two about the implications of the findings.

>>>>> In response to your suggestion, we have expanded the conclusion to include a discussion on the implications of our findings. This addition emphasizes the potential for using our results to inform targeted interventions that address both the psychological and practical aspects of sustainable farming. This revision appears on Page 36, Lines 8 to 12 of the revised manuscript.

“The findings underscore the potential for targeted interventions that leverage psychological insights to foster sustainable farming practices. By understanding the motivational and cognitive processes that influence farmers' behavior, policymakers and practitioners can design more effective strategies that not only promote environmental sustainability but also support the well-being of farmers.”

Reviewer #2: 

1. The study presented offers a comprehensive examination of the psychological drivers influencing farmers' adoption of Net Zero policies, incorporating a wide range of theoretical frameworks. This multi-dimensional approach, encompassing theories such as the Unified Theory of Acceptance and Use of Technology and Self-Determination Theory, enriches our understanding of farmers' attitudes and behaviours towards sustainability initiatives. However, while the integration of diverse psychological perspectives is commendable, it may be beneficial to provide a clearer rationale for the selection of specific theories and how they interact within the context of sustainable agriculture. 

>>>>> This consideration (together with point 4 made by this reviewer below) led to major changes in the structure and positioning of the paper, during which we realized that the psychological theories needed to be organized within a wider context. Therefore, we have separated the frameworks into broadly behavioral adoption—based on existing literature—and purpose-driven, which introduces more value-based considerations. By defining these, we believe the literature is more organized, and the paper makes a more significant contribution to the field. We also feel that this helps cement many aspects of the paper in terms of providing a wider context for the interpretation of the model's findin

---

## [Decision Letter · Decision Letter 1]

8 Aug 2024

Avoiding Silo Approaches : Integrating Psychological Insights into Sustainable Farming

PONE-D-24-11759R1

Dear Dr. Maltby,

We’re pleased to inform you that your manuscript has been judged scientifically suitable for publication and will be formally accepted for publication once it meets all outstanding technical requirements.

Kind regards,

Amar Razzaq, PhD

Academic Editor

PLOS ONE

Additional Editor Comments (optional):

Reviewers' comments:

Reviewer's Responses to Questions

**Comments to the Author**

1. If the authors have adequately addressed your comments raised in a previous round of review and you feel that this manuscript is now acceptable for publication, you may indicate that here to bypass the “Comments to the Author” section, enter your conflict of interest statement in the “Confidential to Editor” section, and submit your "Accept" recommendation.

Reviewer #2: All comments have been addressed

Reviewer #3: All comments have been addressed

2. Is the manuscript technically sound, and do the data support the conclusions?

Reviewer #2: Yes

Reviewer #3: Yes

3. Has the statistical analysis been performed appropriately and rigorously? 

Reviewer #2: Yes

Reviewer #3: Yes

4. Have the authors made all data underlying the findings in their manuscript fully available?

Reviewer #2: Yes

Reviewer #3: Yes

5. Is the manuscript presented in an intelligible fashion and written in standard English?

Reviewer #2: Yes

Reviewer #3: Yes

6. Review Comments to the Author

Reviewer #2: after reviewing your manuscript, I am satisfied with the novelty of the study, interpretation of results and policy of the study.

Reviewer #3: (No Response)

7. PLOS authors have the option to publish the peer review history of their article (what does this mean?). If published, this will include your full peer review and any attached files.

Reviewer #2: **Yes: **Muhammad Waseem

Reviewer #3: No

---

## [Editor Report · Acceptance letter]

4 Oct 2024

PONE-D-24-11759R1 

PLOS ONE

Dear Dr. Maltby, 

I'm pleased to inform you that your manuscript has been deemed suitable for publication in PLOS ONE. Congratulations! Your manuscript is now being handed over to our production team.

Kind regards, 

on behalf of

Associate Professor Amar Razzaq 

Academic Editor

PLOS ONE